# Historical trends in health care-related financial holdings among members of Congress

**Matthew S. McCoy**[1]*, **Matthew Bonci**[2], **Steven Joffe**[1], **Genevieve P. Kanter**[1]

**1** Perelman School of Medicine, University of Pennsylvania, Philadelphia, Pennsylvania, United States of America, **2** University of Zurich and Jacobs Center for Productive Youth Development, Zurich, Switzerland

* mmcco@pennmedicine.upenn.edu

**Data Availability Statement:** All relevant data are within the paper and its Supporting Information files.

**Funding:** The authors received no specific funding for this work.

## Abstract

### Background

Revelations that some members of Congress, including members of key health care committees, hold substantial personal investments in the health care industry have raised concerns about lawmakers' financial conflicts of interest (COI) and their potential impact on health care legislation and oversight.

### Aims

1) To assess historical trends in both the number of legislators holding health care-related assets and the value and composition of those assets. 2) To compare the financial holdings of members of health care-focused committees and subcommittees to those of other members of the House and Senate.

### Methods

We analyzed 11 years of personal financial disclosures by all members of the House and Senate. For each year, we calculated the percentage of members holding a health care-related asset (overall, by party, and by committee); the total value of all assets and health care-related assets held; the mean and median values of assets held per member; and the share of asset values attributable to 9 health asset categories.

### Findings

During the study period, over a third of all members of Congress held health care-related assets. These assets were often substantial, with a median total value per member of over $43,000. Members of health care-focused committees and subcommittees in the House and Senate did not hold health care-related assets at a higher rate than other members of their respective chambers.

**Competing interests:** The authors have declared that no competing interests exist.

## Conclusions

These findings suggest that lawmakers' health care-related COI warrant the same level of attention that has been paid to the COI of other actors in the health care system.

## Introduction

Federal lawmakers affect all aspects of the United States health care system. They shape the way that care is financed and delivered, the way that medical research is conducted, and the way that new drugs and medical devices are approved and marketed. Thus, recent revelations that some members of Congress, including members of key health care committees, hold substantial personal investments in the health care industry have understandably renewed concerns about lawmakers' financial conflicts of interest (COI) and their potential impact on health care legislation and oversight [1–6].

Despite scrutiny from media and watchdog organizations [7], health care-related COIs among members of Congress have rarely been formally studied by academic researchers. Thus, despite having important implications for public health, many aspects of lawmakers' health care-related COI remain poorly understood. A report from *STAT News* found that in 2015, roughly 30% of senators and 20% of representatives held health care-related assets [8]. However, it is not clear how these patterns relate to historical averages, making it difficult to assess whether health care-related COI among members of Congress have increased or decreased over time.

Because every lawmaker has the opportunity to vote on health care-related bills, any lawmaker's financial stake in the industry can raise concerns about COIs and their impact on health care legislation. However, those who sit on key health care committees and subcommittees have far greater involvement in the shaping of health care legislation. Thus, health care industry-related COIs among these members are particularly concerning. A 2009 investigation conducted by the Center for Responsive Politics found that 54 members of House and Senate committees with responsibility for health care legislation had personal financial holding in the health care industry [9]. Yet was not clear from this analysis whether these members held health care-related assets at a higher rate than other members of their respective chambers.

To better understand the scope of health care-related COI among members of Congress, we analyzed 11 years of personal financial disclosures by all members of the House and Senate. Our primary aim was to assess historical trends in both the number of legislators holding health care-related assets and the value and composition of those assets. Our secondary aim was to compare the financial holdings of members of health care-focused committees and subcommittees to those of other members of the House and Senate to determine whether those with the greatest influence over health care legislation and oversight were more likely to have investments in the health care industry.

## Study data and methods

### Data

All members of Congress are required to file annual reports disclosing personal assets. Reports contain information about the source, type, and dollar value of assets (reported in ranges rather than exact figures). While the Office of the Clerk of the House and the Senate Office of Public Records make scanned copies of these reports available as PDFs, they do not provide

aggregate data in a readily analyzable format [10,11]. However, the Center for Responsive Politics (CRP)—a nonpartisan research group that tracks the flow of money in politics—aggregates and digitizes the financial disclosure reports and makes these data publicly available [12].

We used CRP data to obtain personal financial disclosures for members of the House and Senate who submitted reports during the 11-year period from 2004 to 2014 (the most recent year available in bulk data format from CRP at the time of analysis). We identified health sector assets in the CRP data using the protocol described in S1 File. Health care-related asset categories included investments in:

- health care, biotechnology, and life sciences funds;

- pharmaceuticals, biologics, and chemical and molecular diagnostics;

- medical devices, instruments, and supplies;

- health insurance carriers and pharmacy benefit managers;

- pharmacies and wholesale drug distributors;

- health care services, including hospitals, ambulatory care facilities, dialysis facilities, laboratories, and physician practices;

- administrative services, including health care-related information technology services, electronic health record systems, and health care consulting;

- health care-related real estate, including skilled nursing facilities and long-term care facilities

- miscellaneous, including contract research organizations.

We used the public member registry available at Congress.gov to determine House and Senate membership during the study period, and used committee membership lists published by the Government Publishing Office to determine committee and subcommittee membership.

## Study variables

We calculated, for each year, the percentage of members holding a health care-related asset (overall, by party, and by committee); the total value of all assets and health care-related assets held; the mean, 90th percentile winsorized mean (observations in the bottom 5th percentile are replaced with the 5th percentile value and observations in the top 95th percentile are replaced with the 95th percentile value to reduce the effect of outliers), and median values of assets held per member; and the share of asset values attributable to the 9 main asset categories. For comparability of dollar values across years, we deflated all dollar amounts by the Consumer Price Index and reported all amounts in 2019 US dollars.

## Methods

We conducted a descriptive analysis using summary statistics of the study variables. We also conducted graphical analyses, plotting changes of the variables over time.

## Results

Between 2004 and 2014, there were 921 unique members of the House (446 Democrat, 473 Republican, 2 Independent) and 211 unique members of the Senate (112 Democrat, 94 Republican, 5 Independent). Each year during this period, 17%-23% of House members held health care-related related assets (16%-20% of Democrats, 17%-28% of Republicans), compared to 26%-35% of Senate members (21%-38% of Democrats, 27%-41% of Republicans) (**Table 1**).

**Table 1. Lawmakers' health care-related assets in 2004 and 2014.**

| House | | 2004 | 2014 |
|---|---|---|---|
| Members | | 437 | 438 |
| | Democrat | 211 | 207 |
| | Republican | 225 | 230 |
| | Independent | 1 | 1 |
| Members Reporting Any Asset (%) | | 418 (96%) | 365 (83%) |
| Members Reporting Any Health Asset (%) | | 94 (22%) | 79 (18%) |
| | Democrat | 34 (16%) | 40 (19%) |
| | Republican | 60 (27%) | 39 (17%) |
| | Independent | 0 (0%) | 0 (0%) |
| Total Amount of Health Assets Held | | $27.6 million | $46.3 million |
| | Democrat | $7.2 million | $15.2 million |
| | Republican | $20.5 million | $31.1 million |
| | Independent | $0 | $0 |
| Mean Amount of Health Assets Held Per Member (sd) | | $294,036 ($850,692) | $586,477 ($1,330,069) |
| | Democrat | $210,356 ($313,896) | $379,865 ($1,116,383) |
| | Republican | $341,454 ($1,038,883) | $798,385 ($1,503,712) |
| | Independent | $0 (.) | $0 (.) |
| Winsorized (90th pctile) Mean Amount of Health Assets Held Per Member (sd) | | $172,095 ($272,611) | $234,560 ($348,540) |
| | Democrat | $202,087 ($291,491) | $196,131 ($323,993) |
| | Republican | $155,101 ($262,311) | $273,976 ($372,117) |
| | Independent | $0 (.) | $0 (.) |
| Median Amount of Health Assets Held Per Member (iqr) | | $43,311 ($134,664) | $60,480 ($192,770) |
| | Democrat | $43,311 ($345,797) | $43,469 ($128,243) |
| | Republican | $43,649 ($120,793) | $70,196 ($387,692) |
| | Independent | $0 (.) | $0 (.) |
| **Senate** | | 2004 | 2014 |
| Members | | 105 | 108 |
| | Democrat | 48 | 55 |
| | Republican | 56 | 51 |
| | Independent | 1 | 2 |
| Members Reporting Any Asset (%) | | 103 (98%) | 94 (87%) |
| Members Reporting Any Health Asset (%) | | 31 (30%) | 28 (26%) |
| | Democrat | 10 (21%) | 13 (24%) |
| | Republican | 21 (38%) | 14 (27%) |
| | Independent | 0 (0%) | 1 (50%) |
| Total Amount of Health Assets Held | | $6.0 million | $13.2 million |
| | Democrat | $3.2 million | $9.4 million |
| | Republican | $2.8 million | $3.7 million |
| | Independent | $0 | $0.2 million |
| Mean Amount of Health Assets Held Per Member (sd) | | $193,015 ($391,822) | $473,043 ($821,956) |
| | Democrat | $316,088 ($621,788) | $719,429 ($1,053,712) |
| | Republican | $134,409 ($212,432) | $266,165 ($514,061) |
| | Independent | $0 (.) | $166,312 (.) |
| Winsorized (90th pctile) Mean Amount of Health Assets Held Per Member (sd) | | $158,962 ($242,269) | $289,629 ($327,220) |
| | Democrat | $208,944 ($302,856) | $404,240 ($381,828) |

*(Continued)*

**Table 1.**  (Continued)

| House | | | |
|---|---|---|---|
| | Republican | $135,160 ($211,949) | $192,014 ($251,471) |
| | Independent | $0 (.) | $166,312 (.) |
| Median Amount of Health Assets Held Per Member (iqr) | | $60,233 ($154,292) | $165,774 ($399,848) |
| | Democrat | $99,478 ($202,334) | $200,334 ($666,868) |
| | Republican | $60,233 ($109,628) | $110,695 ($271,606) |
| | Independent | $0 (.) | $166,312 (.) |

Fig 1 shows changes over time in health care-related asset holding by party and chamber. In 2004, in both chambers, a higher percentage of Republicans than Democrats held health care-related assets. However, party differences diminished over time. In 2014, there was little difference in health-care related asset holding by party in the House (19% Democrats, 17% Republicans) or Senate (24% Democrats, 27% Republicans).

Among members of Congress reporting at least one health care-related asset in a given year, the total value of legislators' health care-related assets ranged from $34 million (in 2004 and 2005) to $64 million (in 2013). As shown in Fig 2, among members holding health care-related assets, the median total value of health care-related assets per member increased over time from $43,311 for Democrats and $43,986 for Republicans in 2004 to $61,018 for Democrats and $70,196 for Republicans in 2014. However, even as the value of legislators' health care-related assets increased over time, such assets continued to account for a less than 4% of legislators' overall financial holdings.

Legislators held assets in multiple sectors of the health care industry (Fig 3). In 2004, nearly 60% of health care-related assets held by legislators were in pharmaceuticals, biologics, and diagnostics. However, the share of health care-related assets attributable to this sector diminished in subsequent years, reaching a low point in 2011 when the sector accounted for only 20% of all health care-related assets held by members of Congress. During this time, there was

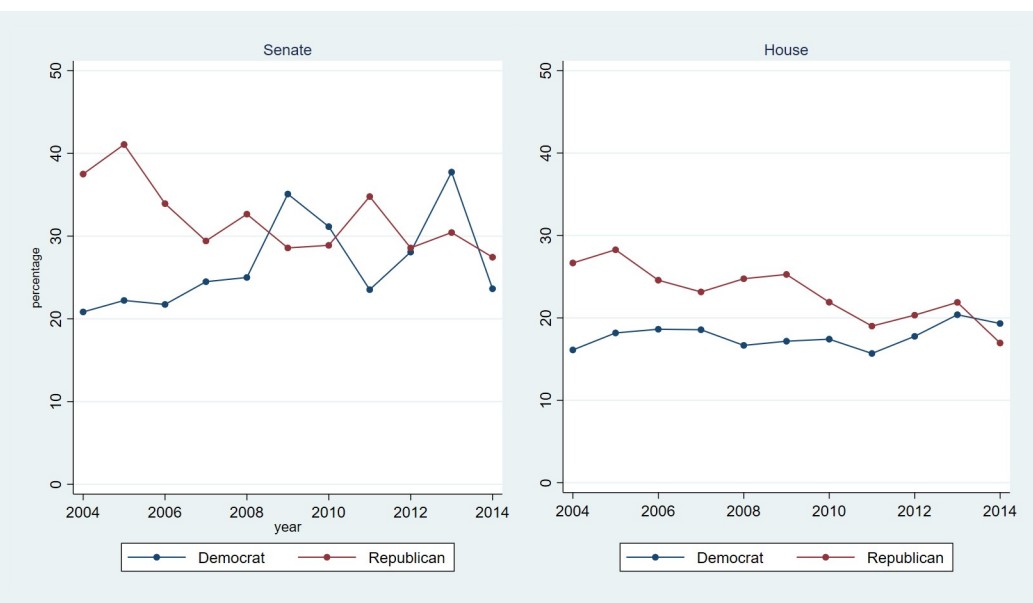

**Fig 1. Share of members holding health care-related assets.**

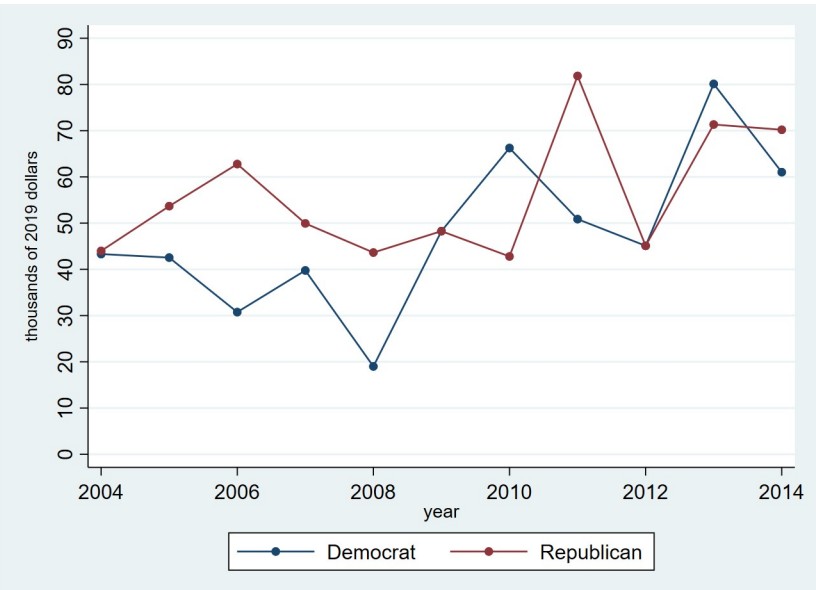

**Fig 2. Median total value of health care-related assets per member per year, by party among those holding at least one health asset in a given year.**

a notable rise in the share of health care-related assets attributable to the health care services sector, which increased by a factor of 7 from 4% in 2004 to a high of 29% in 2011. No other sector accounted from more than 25% of health care-related assets during the study period.

In both the House and Senate, members of committees and subcommittees with responsibility for health care legislation and oversight were no more likely than other members of the same chamber to hold health care-related assets. **Fig 4** compares health care-related asset holding among members of the health care-focused subcommittees of the Energy and Commerce

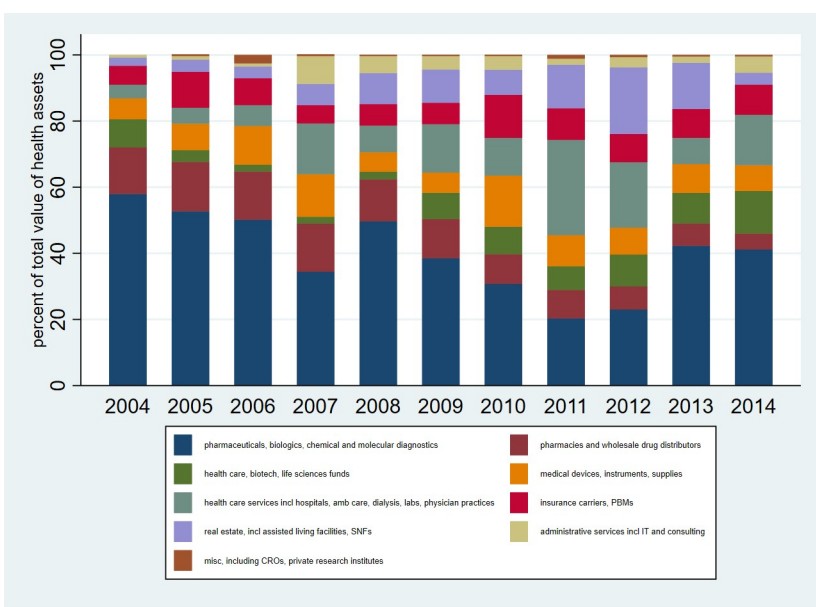

**Fig 3. Share of health care-related asset value attributable to different sectors.**

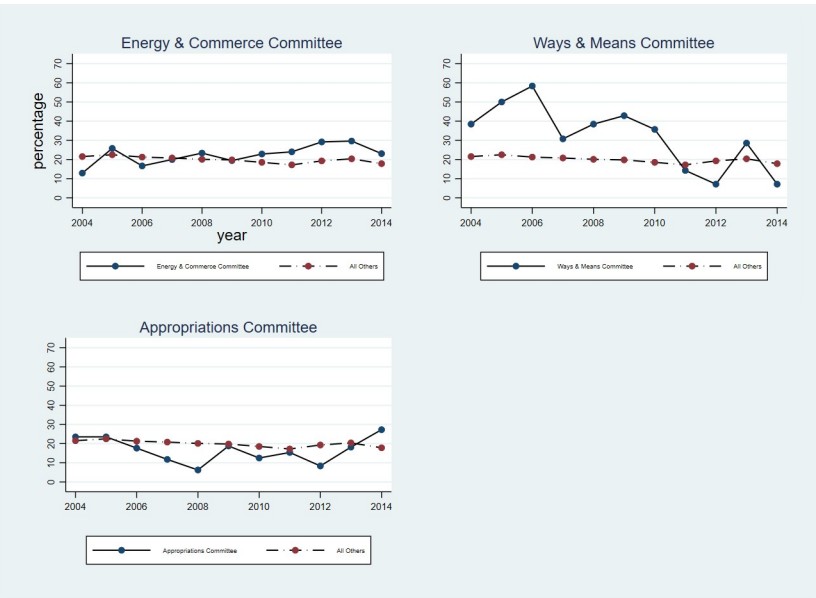

**Fig 4. Share of members holding health care-related assets: House committees.** *All committee samples include only those on health-related subcommittees.

Committee, Ways and Means Committee, and House Appropriations Committee to all other members of the House. **Fig 5** compares health care-related asset holding among members of the Senate HELP Committee and the health care-focused subcommittees of the Finance Committee and Senate Appropriations Committee to all other members of the Senate.

Notably, there was significant year-to-year variation in both the prevalence and value of health care-related asset holding among members of health care-focused committees and

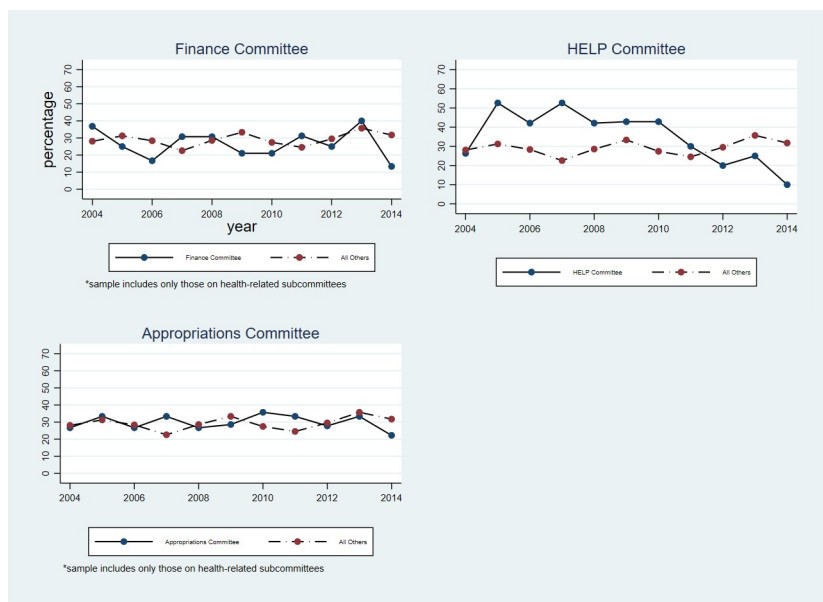

**Fig 5. Share of members holding health care-related assets: Senate committees.** *Finance and Appropriations Committee samples include only those on health-related subcommittees.

subcommittees. For example, in the Health subcommittee of the House Ways and Means Committee, the number of members who held any health care-related asset ranged from a high of 7 in 2006 to a low of 1 in 2012 and 2014, while the median total value of health care-related assets per member ranged from a high of $320,665 in 2010 to a low of $13,170 in 2013. In the Senate HELP Committee, the number of members who held any health care-related asset ranged from a high of 10 in 2005 and 2007 to a low of 2 in 2014, while the median total value of health care-related assets per member ranged from a high of $414,429 in 2014 to a low of $21,656 in 2004.

## Discussion

From 2004–2014—a period of intensive health care legislative activity, including passage of the Affordable Care Act—over a third of all members of Congress held health care-related assets. These assets were often substantial, with a median total value per member of over $43,000. While members of health care-focused committees and subcommittees in the House and Senate did not hold health care-related assets at a higher rate than other members of their respective chambers, multiple members of these committees and subcommittees held hundreds of thousands of dollars of health care-related assets during their committee service. In short, these findings show that at the federal level, health care legislative activities were regularly conducted by lawmakers with significant financial interests in the health care industry.

This study had several limitations. First, we did not have access to current financial disclosure data for members of Congress and are therefore unable to assess trends in health care-related asset holding beyond 2014. This limitation results from shortcomings of financial disclosure reporting. Lawmakers' disclosure forms are often filled out by hand and are published by the House and Senate as individual PDF files rather than in a searchable database. Compiling data on lawmakers' financial disclosures thus requires many hours of labor-intensive abstraction which lags behind the release of disclosure reports. Second, while we analyzed asset holdings in the health care industry, we did not analyze assets in other industries, such as the energy industry, that affect public health. Thus, our findings do not capture the full extent to which lawmakers are invested in industries that have an impact on public health. Finally, we did not analyze the relationship between lawmakers' financial holdings and their legislative activities. Although research on COI in other domains suggests that financial interests can bias professional judgment, we are unable to assess the extent to which health care-related COI might have influenced lawmakers' behavior [13,14].

For over a decade, there has been a great deal of research on the prevalence and impact of COI in medicine [15]. For the most part, however, researchers working in this area have neglected to examine the COI of health policy makers.[12] While federal legislators do not provide patient care or conduct clinical trials, they have a profound impact on the US health care system and, by extension, the health of all Americans. As such, lawmakers' health care-related COI warrant the same level of attention that has been paid to the COI of other actors in the health care system. The study highlights the value of financial disclosure data as a way to better understand the scope of health-care related COI among members of Congress. Drawing on the financial data used in this study, future research should examine the effects of lawmakers' financial interests on health care legislation and oversight.

## Supporting information

**S1 File. Data and methodology.**
(DOCX)

## Author Contributions

**Conceptualization:** Matthew S. McCoy, Steven Joffe, Genevieve P. Kanter.

**Formal analysis:** Matthew Bonci, Genevieve P. Kanter.

**Investigation:** Matthew S. McCoy, Genevieve P. Kanter.

**Methodology:** Matthew S. McCoy, Matthew Bonci, Steven Joffe, Genevieve P. Kanter.

**Project administration:** Matthew S. McCoy, Matthew Bonci, Genevieve P. Kanter.

**Supervision:** Matthew S. McCoy.

**Writing – original draft:** Matthew S. McCoy.

**Writing – review & editing:** Matthew S. McCoy, Matthew Bonci, Steven Joffe, Genevieve P. Kanter.

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
