## [Decision Letter · Decision Letter 0]

9 Jun 2021

Historical Trends in Health Care-Related Financial Holdings Among Members of Congress

PONE-D-21-10335

Dear Dr. McCoy,

We’re pleased to inform you that your manuscript has been judged scientifically suitable for publication and will be formally accepted for publication once it meets all outstanding technical requirements.

Kind regards,

Chang-Qing Gao

Academic Editor

PLOS ONE

Reviewers' comments:

Reviewer's Responses to Questions

**Comments to the Author**

1. Is the manuscript technically sound, and do the data support the conclusions?

Reviewer #1: Yes

2. Has the statistical analysis been performed appropriately and rigorously? 

Reviewer #1: Yes

3. Have the authors made all data underlying the findings in their manuscript fully available?

Reviewer #1: Yes

4. Is the manuscript presented in an intelligible fashion and written in standard English?

Reviewer #1: Yes

5. Review Comments to the Author

Reviewer #1: This is a well designed study that provides useful information on the health sector holding of members of the US Congress. The data will be useful for people who are thinking about the impact of the health care industry on legislation. It would have been interesting to look at the relationship between health care investments and investments in other sectors of the economy as well as income and wealth. I realize the authors did not collect this data, but it would've given a better sense of the influence of the health care industry on Congress as opposed to other industries.

6. PLOS authors have the option to publish the peer review history of their article (what does this mean?). If published, this will include your full peer review and any attached files.

Reviewer #1: **Yes: **David Benjamin Resnik

---

## [Editor Report · Acceptance letter]

28 Jun 2021

PONE-D-21-10335 

Historical Trends in Health Care-Related Financial Holdings Among Members of Congress 

Dear Dr. McCoy:

I'm pleased to inform you that your manuscript has been deemed suitable for publication in PLOS ONE. Congratulations! Your manuscript is now with our production department. 

Kind regards, 

on behalf of

Dr. Chang-Qing Gao 

Academic Editor

PLOS ONE